# COVID-19 Vaccination Willingness and Acceptability in Multiple Sclerosis Patients: A Cross Sectional Study in Iran

**DOI:** 10.3390/vaccines10010135

**Published:** 2022-01-17

**Authors:** Seyed Massood Nabavi, Mehrnoosh Mehrabani, Leila Ghalichi, Mohammad Ali Nahayati, Mehran Ghaffari, Fereshteh Ashtari, Seyed Ehsan Mohammadianinejad, Shahedeh Karimi, Leila Faghani, Sepideh Yazdanbakhsh, Abbas Najafian, Koorosh Shahpasand, Massoud Vosough

**Affiliations:** 1Arya Group for Treatment and Research in Multiple Sclerosis, Tehran 1435864464, Iran; mehrnooshmehrabani@yahoo.com (M.M.); Ghalichi.l@iums.ac.ir (L.G.); karimi58shah@gmail.com (S.K.); Leyla_Faghani@yahoo.com (L.F.); S_yazdanbakhsh@yahoo.com (S.Y.); ramin116@yahoo.com (A.N.); 2Department of Regenerative Medicine, Cell Science Research Center, Royan Institute for Stem Cell Biology and Technology, Tehran 1665659911, Iran; 3Mental Health Research Center, Psychosocial Health Research Institute, Iran University of Medical Sciences, Tehran 1449614535, Iran; 4Neurology Department, Mashhad University of Medical Sciences, Mashhad 9177948564, Iran; nahayatima@gmail.com; 5Neurology Department, Shahid Beheshti University of Medical Sciences, Tehran 1983969411, Iran; m.ghaffari@sbmu.ac.ir; 6Neurology Department, Isfahan University of Medical Sciences, Isfahan 73461-81746, Iran; Fashtari231@gmail.com; 7Neurology Department, Jondishapour University of Medical Sciences, Ahvaz 15794-61357, Iran; Ehsanneuro@gmail.com; 8Department of Brain and Cognitive Sciences, Cell Science Research Center, Royan Institute for Stem Cell Biology and Technology, Tehran 1665659911, Iran; shahpasand09@gmail.com

**Keywords:** COVID-19 vaccination, multiple sclerosis, vaccine willingness, vaccine acceptability, vaccine hesitancy

## Abstract

Multiple sclerosis (MS) is a chronic, predominantly immune-mediated degenerative disease of the central nervous system. Due to prolonged use of immunomodulatory and immunosuppressive medications, vaccine hesitancy could be common among MS patients. Our main aim in the current study was to evaluate the willingness and acceptability of COVID-19 vaccination in patients with MS. In our multicenter cross-sectional questionnaire-based clinical study, 892 patients completed the questionnaire between May to June 2021. The questionnaire consisted of demographic data, MS disease-related factors, history of COVID-19 infection/vaccination, and any existing comorbidities. Statistical analysis was performed using SPSS software version 19. Overall, 68% of the participants expressed willingness to be vaccinated. Major causes of vaccine refusal in our patients were the fear of reducing the efficacy of disease modifying drugs (DMDs) upon vaccination as well as distrusting the vaccines and overestimation bias in the power of their innate immunity and potential COVID-19 resistance. Some demographic factors affected vaccination enthusiasm in our study. Our findings did not show significant correlation between the age and comorbidity and vaccine willingness. Only one-third of our patients received their vaccine information from healthcare providers. The majority of them received these data from official broadcasting channels and social media. However, despite several concerns, the willingness of COVD-19 vaccination in the Iranian MS patients is remarkable.

## 1. Introduction

Late December 2019, the world faced a pandemic infectious disease, caused by severe acute respiratory syndrome coronavirus 2 (SARS CoV-2). The spread of the coronavirus disease 2019 (COVID-19) caused significant morbidity and mortality as well as substantial psychosocial complications and economic crisis all over the world [1].

Novel therapeutic approaches for the treatment of COVID-19, included passive immunotherapy, cell-based therapies (including immune cell and non-immune cell therapies), gene-editing-based therapeutics [2], monoclonal antibodies, and anti-viral drugs [3]. Vaccination has been among the most effective and cost-beneficial interventions to control this pandemic [4]. Up to now, 56.8% of the world population has received at least one COVID-19 vaccine dose, and 43.3% have been fully vaccinated. A total of 8.7 billion doses have been administered globally, and 33.87 million shots are currently being administered daily (https://ourworldindata.org/covid-vaccinations (accessed on 15 November 2021)). Unfortunately, only 7.6% of people in low-income countries have received at least one dose [5].

According to a systematic review conducted in Asella, Ethiopia, factors such as age, educational status, gender, income, occupation, marital status, race/ethnicity, perceived risk of COVID-19, trust in healthcare system, health insurance, attitude towards vaccine, perceived benefit of vaccine, perceived vaccine barriers, self-efficacy, being up-to-date on vaccinations, having tested for COVID-19 in the past, perceived efficacy of the COVID-19 vaccination, recommended for vaccination, political leaning, perceived severity of COVID-19, perceived effectiveness of COVID-19 vaccine, belief that vaccination makes them feel less worried about COVID-19, believing in mandatory COVID-19 vaccination, perceived potential vaccine harms, presence of chronic disease, confidence, COVID-19 vaccine safety concern, working in healthcare field, believing vaccines can stop the pandemic, fear about COVID-19, cues to action, COVID-19 vaccine hesitancy, complacency, and receiving any vaccine in the past 5 years were correlated to the patient’s willingness to receiving COVID-19 vaccine [6]. 

However, vaccine hesitancy in some societies has not allowed health workers to vaccinate entire populations against COVID-19. Vaccine hesitancy is defined as “delay in acceptance of vaccination or refusal of vaccination despite availability” [5,7]. The rapid development of COVID-19 vaccines and emergency authorization of the approved vaccine has raised concerns about the safety of such vaccines, leading to vaccine hesitancy [7]. Vaccine hesitancy is initiated from various factors, such as lack of knowledge and perceived risks of vaccination. Common concerns associated with the hesitancy encompass a broad spectrum, including skepticism on its safety, effectiveness, and under estimation of infection as a life-threatening factor.

MS as a chronic immune-mediated degenerative disease, has worldwide prevalence with 2.8 million cases annually [8]. This population might be at an increased risk of COVID-19 complications and some of these patients have had contracted severe COVID-19 disease. According to several reports, the risk of COVID-19 infection is not greater than the general population in MS patients, however older age, disability, presence of some comorbidities, and also the use of some specific DMDs (disease modifying drugs), can increase the risk of hospitalization, ICU admission, and probably a higher death rate in MS patients [9]. Among MS patients, because of their regular use of immunomodulatory and immunosuppressive medications, the concern about the inefficacy of the vaccines, any adverse effects on their disease course, and possible adverse impact or interference with their regular medications are significant factors in vaccine hesitancy [10]. However, experts have recommended that MS patients, including those taking one of more than 20 DMDs approved by the FDA, should receive the vaccine, when offered. However, certain DMDs, including Ocrelizumab and Fingolimod, may reduce the efficacy of COVID-19 vaccines [11]. Since vaccination has been approved as an effective and acceptable approach for controlling COVID-19 [12,13] and also as infectious diseases may increase the risk of disease progression in MS patients and result in reduced life expectancy [14]. COVID-19 vaccination has started in Iran since May 2021 [15]. According to COVID-19 vaccination guidelines in Iran [16], elderly people and patients suffering from certain chronic diseases such as multiple sclerosis (MS) are prioritized.

So far, there have been five studies describing MS patient’s willingness for COVID-19 vaccination, two studies in Europe and the UK, and three studies in the USA. In the mentioned studies, between 66.3 up to 94.4% of patients showed enthusiasm for vaccination. These studies showed that some demographic (age, sex, etc.) and disease-related factors could affect the vaccination willingness and result in vaccine hesitancy [10,12,13,17,18].

Nonetheless, remarkable differences in vaccine acceptance rates can be observed across different countries and subpopulations, supporting the underlying complex and unpredictable interplay among demographic, geopolitical, and cultural aspects [19].

This study was performed at the beginning of COVID-19 vaccination process in Iran to assess the vaccination willingness and acceptability in Iranian MS patients.

## 2. Methods

This is a multicenter cross-sectional questionnaire-based clinical study, which was performed between May to June 2021, in Iran. We included all the patients diagnosed with MS, visited by neurologists in five distinct neurology clinics in five different cities between May to June 2021. The total number of the patients who consented to participate in this cohort were 892. The total number of all visited MS patients in these clinics were 1200. The patients were initially informed and completed the questionnaire consent. The anonymous questionnaires were designed according to the ethical guidelines and on the first page. All participants were informed about the study and were voluntarily enrolled.

The questionnaire consisted of demographic data (age, gender, race, marital status, and education level), MS disease-related factors including disease duration, course, current disease-modifying therapy, disability status by expanded disability status scale (EDSS), history of COVID-19 infection/vaccination, and any existing comorbidities. We also inquired their perception of disease risk and death risk, as well as their trust and willingness in different types of vaccines.

Statistical analysis was performed using SPSS software version 19. Descriptive statistical methods were administered. Chi-square test and t-test were applied to compare categorical and continuous variables between subgroups. *p* values less than 0.05 were considered statistically significant. Finally, we included variables with strong univariate association (*p* value ≤ 0.20) in the multivariable logistic regression model.

## 3. Results

Overall, 892 cases completed the questionnaire; among them 76% were female. The age of participants ranged from 16 to 64 (mean age 37.8, SD: 9.22 years). The type of disease was progressive in 18% and relapsing in 82% of cases. Participants live in 80 different locations, mainly in three major cities. Based on their disability status, 79% were categorized as patients with no or minimal disability (EDSS: 0–3.5), 15% as mild to moderate (EDSS: 4–5.5), and 6% as severe (EDSS ≥ 6). The mean duration of their disease was 9 years (SD: 6.51) and 111 patients (12%) of them reported a concurrent comorbidity.

Among all cases, 190 (21%) reported a history of COVID-19 infection during the pandemic, 545 (61%) had been vaccinated with at least one dosage, and 478 had received both shots. The neurologists of 51% of the cases reported COVID-19 complications during their visits. Almost 98% of the vaccinated cases had received an inactivated virus vaccine type, Sinopharm (BBIBP-CorV). Sinopharm BBIBP-CorV, is a 2-dose β-propiolactone-inactivated, aluminium hydroxide-adjuvanted vaccine administered on a 0/21–28-day schedule which seems a safe choice for MS patients [20]. Only one participant refused a second dose due to side effects from the first dose.

Overall, 68% of participants expressed willingness to be vaccinated. Males showed higher vaccine willingness compared to females (91% vs. 85%, *p* value = 0.02). The educated and employed patients showed more enthusiasm for vaccination. History of COVID-19 infection, concurrent comorbidities, and type of DMDs, disease type (relapsing or progressive), or the level of disabilities had no effect on willingness (Table 1). The vaccine willingness rate was not associated with either age or disease duration (*p* value: 0.63 and 0.06, respectively) in the patients.

The majority of the participants had higher risk perception for COVID-19 infection (70%) and death (55%); considering the type of MS disease and the prescribed medications. In the patients who considered COVID-19 as a life-threatening infection and the patients who had willingness for vaccination, the risk perception was also high (*p* value < 0.001 for both). Considering the manufacturing country of the vaccines, 65% had trusted in Iranian vaccines, 83% had trust in vaccines manufactured in other countries, and 12% did not believe in vaccination. Only 8.38% of the latter group wished to be vaccinated (*p* value < 0.001).

Almost 40% of the 268 participants who declared some reluctance against vaccination, were concerned about the probable interaction of the vaccine with their MS medications. Other reasons included vaccine inefficiency (25%), MS deterioration upon vaccination (16%), innate resistance (16%), and lack of belief in the COVID-19 disease (3%).

Multivariable logistic regression model was applied for predicting vaccine willingness, including sex, marital status, MS type, DMF, university education, and having a paid job as variables. Finally, three variables (sex, university education, and marital status) were included and presented in Table 2.

## 4. Discussion

We herein performed a cross-sectional multicenter study to assess the willingness and acceptability of COVID-19 vaccination in the Iranian MS population, between May to June 2021. According to our findings, 86% of Iranian MS patients were willing to receive a COVID-19 vaccine. It has been reported that several demographic and disease-related factors can affect COVID-19 vaccination willingness. In our study, subjects were almost close to the general population in the United States and were similar to previous studies [11,13,17,21,22,23,24].

Some national surveys in other countries, with sample sizes of 316–7632, that were done between April and August 2020 found that 49–69% of participants intended to be vaccinated, 17–32% were not sure, and 11–35% did not want to be vaccinated. The recent study of vaccine willingness among MS patients found that 66% of participants were willing and 15.4% were unwilling to receive a COVID-19 vaccine [13].

A cross-sectional study was conducted among adults in Saudi Arabia between January and March 2021 to determine their willingness to receive a COVID-19 vaccine. A descriptive analysis of the 531 participants showed that 61.8% were willing to get the COVID-19 vaccine, while 38.2% were not. COVID-19 vaccine hesitancy was more among women (44.9%), those 34–49 years of age (47.9%), those who were married (41.9%), employed (39.7%), had lower educational attainment (40%), and were urban dwellers (40.8%) [25].

Some demographic factors affected vaccination enthusiasm in our study. However, age, disease duration, disability status, comorbidity, DMD type (first line and second line), and disease type (relapsing/remitting vs. progressive) did not significantly affect vaccine acceptance. In one study, vaccine willingness was association with the age and disability status of patients [17]. In three other studies, source of information and especially, the given information by neurologists affected vaccine willingness in MS patients, which is inconsistent with our findings. However, education and risk perception for COVID-19, affected vaccine willingness, which is consistent with previous findings. We also realized that outdoor working had a significant effect on vaccination enthusiasm.

Major causes of vaccine refusal in our patients was the fear of reducing the efficacy of DMDs upon vaccination as well as distrusting the vaccines. 

Only one third (31.6%) of the patients in our study received vaccine information from healthcare providers. The majority of our patients received information from official broadcasting channels and social media.

In contrast to the European study, our findings did not show significant correlation between the age and comorbidity and vaccine willingness, which is consistent with the American study [13].

On the other hand, Boekel L. et al., published a commentary in which they found that the proportion of patients with autoimmune diseases and controls who would be willing to get vaccinated against SARS-CoV-2 was similar. In that study, male participants and individuals older than 60 years were both approximately twice as likely to be willing to get vaccinated. The most common reasons for refusing or doubting vaccination in both patients and controls were concerns for possible adverse events. In addition, there is no long-term study on the side effects of COVID-19 vaccines [26].

The strength of our study is the remarkable number of enrolled subjects who entered the study, and they were from different regions of the country, with various demographic factors such as: Broad ranges of age, sex, disability status, and different types of DMDs.

The main disadvantage of our study was excluding the patients who did not seek medical attention (due to advanced disability or fear of COVID-19 infection) since we only enrolled patients who were visited in the clinics.

## 5. Conclusions

Although there are several concerns against COVID-19 vaccination, the willingness of COVD-19 vaccination in the Iranian MS patients is remarkable. The mentioned concerns were from uncertainty in the vaccines’ effectiveness, probable adverse effects, and possibility of interference with DMDs, and even a lack of access to sufficient information. This willingness demonstrates the patients’ awareness and enlightenment about the effectiveness of vaccination in controlling the global pandemic. All health experts should inform patients regarding the advantages of COVID-19 vaccination. Finally, it should be mentioned that some concerns could be vaccine dependent. 

## Figures and Tables

**Table 1 vaccines-10-00135-t001:** Distribution of vaccine willingness based on characteristics of Multiple Sclerosis patients.

		Low		High		*p* Value
**Sex**	Male	16	8.6%	170	91.4%	0.12
	Female	102	15.2%	571	84.8%	
**Marital status**	Married	68	12.3%	487	87.7%	0.07
	Single	50	16.7	250	83.3%	
**Type**	Relapsing	92	13.0%	613	87.0%	0.15
	Progressive	26	17.6%	122	82.4%	
**EDSS**	0–3.5	92	13.6%	585	86.4%	0.77
	4–5.5	18	13.8%	112	86.2	
	≥6	8	17.4%	38	82.6%	
**DMF**	No DMF	12	22.6%	41	77.4%	0.08
	DMF 1	52	12.0%	381	88%	
	DMF 2	40	14.9%	229	85.1%	
**University education**	No	61	17.3%	292	82.7%	0.02
	Yes	58	11.5%	445	88.5%	
**Paid job**	No	83	15.3%	461	84.7%	0.03
	Yes	30	9.9%	274	90.1	
**COVID History**	No	91	14.6%	533	85.4%	0.87
	Yes	26	14.1%	158	85.9%	
**Comorbidity**	No	94	13.4%	607	86.6%	0.23
	yes	19	17.8%	88	82.2%	
**Perception of vaccine risk**	No	62	24.5%	191	75.5%	<0.001
	Yes	54	9.2%	532	90.8%	
**Perception of death risk**	No	82	22.1%	289	77.9%	<0.001
	yes	33	7.2%	427	92.8%	
**Concerns about the vaccine**	Interaction with MS medications	18	16.5%	91	83.5%	<0.001
	Vaccines inefficiency	45	67.2%	22	32.8%	
	Disease deterioration after vaccination	16	41.0%	23	59.0%	
	Innate resistance	13	32.5%	27	67.5%	
	Not believing in COVID-19	5	62.%%	3	37.5%	

DMD: Disease modifying drugs; EDSS: Expanded disability status scale.

**Table 2 vaccines-10-00135-t002:** Logistic regression model for variables predicting vaccine willingness.

	OR	95% CI	*p* Value
Male gender	2.1	1.2–3.8	0.015
University education level	1.6	1.1–2.5	0.021
Marital status, married	1.5	1.0–2.3	0.048

CI: Confidence interval; OR: Odds ratio.

## Data Availability

The data presented in this study are available on request from the corresponding author. The data are not publicly available due to privacy restrictions. All data that were analyzed during this study are included in this published article.

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
