# Peer review of "COVID-19 Vaccination Willingness and Acceptability in Multiple Sclerosis Patients: A Cross Sectional Study in Iran"

_vaccines, 2022, doi:10.3390/vaccines10010135_

Round 1
Reviewer 1 Report
In this second round of revision, the manuscript submitted by Nabavi et al. entitled "COVID-19 vaccination willingness and acceptability in Multiple Sclerosis patients: A cross sectional study in Iran" is improved regarding the quality of the English language. In addition, more data was also added and the discussion amended. Thus, the opinion of this reviewer is that the article can be suitable for the publication at the present form.
Author Response
Thank you for your positive and promising feedback
Reviewer 2 Report
Estimated Authors,
I've appreciated your considerable efforts to cope with my previous recommendations. In fact, nearly all my previous concerns have been either properly addressed or solved by the amendments you've performed.
Only a minor post-acceptance recommendation:
the caption of the rows from Table 2 should be refined. More precisely, according to main Table 1:
MALE --> either "male gender" of "male sex"
MARRIED --> "marital status, married"
UNIVERSITY EDUCATION --> "UNIVERSITY LEVEL EDUCATION" of "UNIVERSITY LEVEL EDUCATION - YES"
I repeat, such fixing may be performed as a post-acceptance editing. Therefore, I'm endorsing the acceptance of this paper.
Author Response
Thank you for your positive and promising feedback.
The mentioned points were modified accordingly.

This manuscript is a resubmission of an earlier submission. The following is a list of the peer review reports and author responses from that submission.
Round 1
Reviewer 1 Report
General recommendations
References must be prepared according to the journal's rules, journal’s name must be in italics, not title.
Introduction
The first paragraphs in the Introduction section contains information that is overly general and were published in a scientific article. It would be best to revise introduction section to describe the importance of the studied topic – vaccination acceptability in Multiple Sclerosis patients.
Materials and Methods.
For questionnaire-based studies it is recommended to provide the questionnaire as supplementary material.
Please add information about the ethical approval for this study (No and date of protocol).
Results
In this study, you indicate that only one participant refused second dose due to side effects of the first dose. Are side effects of vaccine were recorded in this study?
If yes, please give this information as separate table.
References
Please correct all references
Author Response
References must be prepared according to the journal's rules, journal’s name must be in italics, not title.
Answer: Thank you for your comment. To address your concern, references have been corrected according to the journal's rules.
Introduction:
The first paragraphs in the Introduction section contains information that is overly general and were published in a scientific article. It would be best to revise introduction section to describe the importance of the studied topic – vaccination acceptability in Multiple Sclerosis patients.
Answer: Thank you for your comment. To address your concern, some sentences in the introduction has been deleted and changed and also a paragraph has been added in lines 74 to 80.
Materials and Methods.
For questionnaire-based studies it is recommended to provide the questionnaire as supplementary material.
Answer: Thank you for your comment. To address your concern, questionnaire was included as an attachment.
Please add information about the ethical approval for this study (No and date of protocol).
Answer: Thank you for your comment. It was a questionnaire-based study and we informed all participants about the study and they voluntarily accepted to participate. Also, the questionnaires were anonymous and the participants approved publication of the data. We did not have any intervention, and according to rules in IRAN, there was no need for the ethical approval.
Results:
In this study, you indicate that only one participant refused second dose due to side effects of the first dose. Are side effects of vaccine were recorded in this study?
If yes, please give this information as separate table.
Answer: Thank you for your comment. No, we are going to report the side effects in another study.
References
Please correct all references
Answer: Thank you for your comment. To address your concern, all the references have been corrected.

Reviewer 2 Report
The manuscript submitted by Nabavi et al., entitled "COVID-19 vaccination willingness and acceptability in Multiple Sclerosis patients: A cross sectional study in Iran" aims to evaluate the willingness and acceptability of COVID-19 vaccination in patients with Multiple Sclerosis. For that, authors used a questionnaire completed by 892 patients between May to June 2021.
Overall 86% of the participants expressed willingness to be vaccinated, with the major cause of vaccine refusal being the fear of reducing the efficacy of DMDs upon vaccination.
The manuscript needs to be read by an English native speaker and the scarce results provided are not new, thus not supporting its publication in a journal like "Vaccines".
The manuscript is just an Epidemiologic Study, in Iran, with no hypothesis-driven. Indeed the work only reflects a very rudimentary statistical study.
Author Response
The manuscript needs to be read by an English native speaker and the scarce results provided are not new, thus not supporting its publication in a journal like "Vaccines".
Answer: Thank you for your comment. To address your concern, the manuscript edited by an English expert.
The manuscript is just an Epidemiologic Study, in Iran, with no hypothesis-driven. Indeed the work only reflects a very rudimentary statistical study.
Answer: Thank you for your comment. It should be mentioned that, other similar studies have reported epidemiologic cross-sectional descriptive data as our study, and our hypothesis was that some MS patients would refuse to receive COVID-19 vaccine due to different reasons. In addition, in the updated version, we included more data and tables to address this concern.
Reviewer 3 Report
Estimated Authors of the paper "COVID-19 vaccination willingness and acceptability in Multiple Sclerosis patients: A cross sectional study in Iran",
I've been asked to review the present study for Vaccines. In this original article, the study group lead by Dr. NABAVI has assessed knowledge, attitudes and practices of 118 Iranian individuals affected by MS in various stages of severity (EDSS 0-3.5 86.4%) towards SARS-CoV-2 vaccines. In a logistic regression model, male gender (OR 2.1, 95%CI 1.2 to 3.8), higher education status (OR 1.6, 95%CI 1.1 to 2.5), and being married (OR 1.5, 95%CI 1.0 to 1.3) were identified as main effector variables towards the outcome variable of being favourable towards SARS-CoV-2 vaccine.
The study is both interesting and consistent with the aims of VACCINES, but several improvements are required before its acceptance, and namely:
a) please include further details on the selection process; were the MS patients recruited as a convenience sample (i.e. all consecutive patients from the contributing centers)? or did Authors opted in for a more appropriate sampling strategy? anyway, it must be reported in details.
b) according to the health belief model, risk perception may be a significant effector of vaccine acceptance; therefore, it is very important to include in the data presentation some information about the risk perception of study participants. According to the main text "Majority of the participants had higher risk perception for COVID-19 infection (70%) and death (55%)". This is very interesting, but I was unable to retrieve such information from Table 1. Similarly, other interesting information (e.g. "Almost 40% of the 268 participants, who declared some reluctance against vaccination, concerned about the probable interaction of vaccine with their MS medications. Other reasons included vaccines inefficiency (25%), disease deterioration upon vaccination (16%), innate resistance (16%), and COVID-19 disease (3%)") have been only marginally reported in the methods. Please reframe your text with a more detailed data reporting.
c) "Finally, logistic regression method was applied" is a statement inappropriate. Authors must describe in more appropriate terms the model they defined, which outcome and effector variables, as well as covariates were included, etc.
d) Authors must more accurately double check their English text. Unfortunately, several significant typos and jargon still remains. For example: "Vaccine hesitancy, defined as “delay in acceptance coincident with or refusal of vaccination despite availability” --> full stop.
Author Response
- a) please include further details on the selection process; were the MS patients recruited as a convenience sample (i.e. all consecutive patients from the contributing centers)? or did Authors opted in for a more appropriate sampling strategy? anyway, it must be reported in details.
Answer: a) We included all patients diagnosed with MS, visited by neurologists in five neurology clinics in five cities between May to June 2021. The total number of the patients who consented to participate in this cohort was 892. The total number of all visited MS patients in these clinics was 1200.
- b) according to the health belief model, risk perception may be a significant effector of vaccine acceptance; therefore, it is very important to include in the data presentation some information about the risk perception of study participants. According to the main text "Majority of the participants had higher risk perception for COVID-19 infection (70%) and death (55%)". This is very interesting, but I was unable to retrieve such information from Table 1. Similarly, other interesting information (e.g. "Almost 40% of the 268 participants, who declared some reluctance against vaccination, concerned about the probable interaction of vaccine with their MS medications. Other reasons included vaccines inefficiency (25%), disease deterioration upon vaccination (16%), innate resistance (16%), and COVID-19 disease (3%)") have been only marginally reported in the methods. Please reframe your text with a more detailed data reporting.
Answer: b) Thank you for your comment. To address your comment, we replaced the table 1 as bellow: (It is highlighted in lines 160-161)
|
|
|
low |
high |
P value |
||
|
sex |
Male |
16 |
8.6% |
170 |
91.4% |
0.12 |
|
female |
102 |
15.2% |
571 |
84.8% |
||
|
Marital status |
married |
68 |
12.3% |
487 |
87.7% |
0.07 |
|
single |
50 |
16.7 |
250 |
83.3% |
||
|
type |
relapsing |
92 |
13.0% |
613 |
87.0% |
0.15 |
|
progressive |
26 |
17.6% |
122 |
82.4% |
||
|
EDSS |
0 – 3.5 |
92 |
13.6% |
585 |
86.4% |
0.77 |
|
4 – 5.5 |
18 |
13.8% |
112 |
86.2 |
||
|
³6 |
8 |
17.4% |
38 |
82.6% |
||
|
DMF |
No DMF |
12 |
22.6% |
41 |
77.4% |
0.08 |
|
DMF 1 |
52 |
12.0% |
381 |
88% |
||
|
DMF 2 |
40 |
14.9% |
229 |
85.1% |
||
|
University education |
No |
61 |
17.3% |
292 |
82.7% |
0.02 |
|
Yes |
58 |
11.5% |
445 |
88.5% |
||
|
Paid job |
No |
83 |
15.3% |
461 |
84.7% |
0.03 |
|
Yes |
30 |
9.9% |
274 |
90.1 |
||
|
COVID History |
No |
91 |
14.6% |
533 |
85.4% |
0.87 |
|
Yes |
26 |
14.1% |
158 |
85.9% |
||
|
comorbidity |
No |
94 |
13.4% |
607 |
86.6% |
0.23 |
|
yes |
19 |
17.8% |
88 |
82.2% |
||
|
Perception of vaccine risk |
No |
62 |
24.5% |
191 |
75.5% |
<0.001 |
|
yes |
54 |
9.2% |
532 |
90.8% |
||
|
Perception of death risk |
No |
82 |
22.1% |
289 |
77.9% |
<0.001 |
|
yes |
33 |
7.2% |
427 |
92.8% |
||
|
Concerns about the vaccine |
Interaction with MS medications |
18 |
16.5% |
91 |
83.5% |
<0.001 |
|
Vaccines inefficiency |
45 |
67.2% |
22 |
32.8% |
||
|
Disease deterioration after vaccination |
16 |
41.0% |
23 |
59.0% |
||
|
Innate resistance |
13 |
32.5% |
27 |
67.5% |
||
|
Not believing in COVID-19 |
5 |
62.%% |
3 |
37.5% |
||
- c) "Finally, logistic regression method was applied" is a statement inappropriate. Authors must describe in more appropriate terms the model they defined, which outcome and effector variables, as well as covariates were included, etc.
Answer: c) Thank you for your comment. To address this comment, new sentences have been added to the text and highlighted in methods (lines 120-121) and results (lines 156-159).
- d) Authors must more accurately double check their English text. Unfortunately, several significant typos and jargon still remains. For example: "Vaccine hesitancy, defined as “delay in acceptance coincident with or refusal of vaccination despite availability” --> full stop.
Answer: d) Thank you for your comment. To address this concern, This issue corrected and highlighted in the text in lines 67 and 68. Also the text revised by an English expert.